# Pre-Clinical Evaluation of the Proteasome Inhibitor Ixazomib against Bortezomib-Resistant Leukemia Cells and Primary Acute Leukemia Cells

**DOI:** 10.3390/cells10030665

**Published:** 2021-03-17

**Authors:** Margot S.F. Roeten, Johan van Meerloo, Zinia J. Kwidama, Giovanna ter Huizen, Wouter H. Segerink, Sonja Zweegman, Gertjan J.L. Kaspers, Gerrit Jansen, Jacqueline Cloos

**Affiliations:** 1Cancer Center Amsterdam, Department of Hematology, Amsterdam UMC, Vrije Universiteit, 1081 HV Amsterdam, The Netherlands; m.roeten@amsterdamumc.nl (M.S.F.R.); j.vanmeerloo@amsterdamumc.nl (J.v.M.); z.kwidama@amsterdamumc.nl (Z.J.K.); giohzn@gmail.com (G.t.H.); w.segerink@hotmail.com (W.H.S.); s.zweegman@amsterdamumc.nl (S.Z.); 2Princess Maxima Center of Pediatric Oncology, 3584 CS Utrecht, The Netherlands; gjl.kaspers@amsterdamumc.nl; 3Emma Children’s Hospital, Amsterdam UMC, Vrije Universiteit Amsterdam, Pediatric Oncology, 1105 AZ Amsterdam, The Netherlands; 4Amsterdam Rheumatology and Immunology Center, Cancer Center Amsterdam, Amsterdam UMC, Vrije Universiteit, 1081 HV Amsterdam, The Netherlands; g.jansen@amsterdamumc.nl

**Keywords:** leukemia, proteasome, proteasome inhibitor, ixazomib, BTZ resistance, drug resistance

## Abstract

At present, 20–30% of children with acute leukemia still relapse from current chemotherapy protocols, underscoring the unmet need for new treatment options, such as proteasome inhibition. Ixazomib (IXA) is an orally available proteasome inhibitor, with an improved safety profile compared to Bortezomib (BTZ). The mechanism of action (proteasome subunit inhibition, apoptosis induction) and growth inhibitory potential of IXA vs. BTZ were tested in vitro in human (BTZ-resistant) leukemia cell lines. Ex vivo activity of IXA vs. BTZ was analyzed in 15 acute lymphoblastic leukemia (ALL) and 9 acute myeloid leukemia (AML) primary pediatric patient samples. BTZ demonstrated more potent inhibitory effects on constitutive β5 and immunoproteasome β5i proteasome subunit activity; however, IXA more potently inhibited β1i subunit than BTZ (70% vs. 29% at 2.5 nM). In ALL/AML cell lines, IXA conveyed 50% growth inhibition at low nanomolar concentrations, but was ~10-fold less potent than BTZ. BTZ-resistant cells (150–160 fold) displayed similar (100-fold) cross-resistance to IXA. Finally, IXA and BTZ exhibited anti-leukemic effects for primary ex vivo ALL and AML cells; mean LC_50_ (nM) for IXA: 24 ± 11 and 30 ± 8, respectively, and mean LC_50_ for BTZ: 4.5 ± 1 and 11 ± 4, respectively. IXA has overlapping mechanisms of action with BTZ and showed anti-leukemic activity in primary leukemic cells, encouraging further pre-clinical in vivo evaluation.

## 1. Introduction

Acute leukemia is the most common type of pediatric malignancy, accounting for almost 30% of all cancer cases in this age group [1]. While the 5-year survival rates of acute lymphoblastic leukemia (ALL) and acute myeloid leukemia (AML) are 83–94% and 65–70%, respectively [2,3,4,5], 20–30% of the patients still relapse from current treatment [2,4,6]. Therefore, there is an unmet need to find new drugs or drug combinations to improve the outcome of pediatric leukemia patients.

Building upon the successful application of the proteasome inhibitor (PI) Bortezomib (BTZ) in the treatment of multiple myeloma (MM) [7,8,9,10] and some solid tumors [11], BTZ has also been proposed for the treatment of acute leukemia [12,13,14]. BTZ has already been evaluated in pediatric acute leukemia, either alone or in combination with other common anti-leukemic drugs such as dexamethasone, vincristine, doxorubicin, and asparaginase. These clinical trials mainly involved relapsed ALL and showed promising results with complete remission rates up to 80%, without excessive toxicity, but recurrences occur [14,15,16,17,18,19]. For pediatric AML, a recent study revealed that BTZ in combination with standard chemotherapy did not improve treatment outcome [20]. These observations, as well as the development of resistance to BTZ and the fact that peripheral neuropathy (PN) occurs which hampers treatment [17,21,22,23], provided a rationale for the development of proteasome inhibitors (PIs) which could overcome these limitations [6,23,24].

Ixazomib (IXA) is an orally bioavailable drug with an improved pharmacokinetic profile (longer plasma half-life than BTZ) and safety profile (reduction in PN) [25,26,27,28]. IXA shares the property of BTZ of preferential (reversible) binding to the proteasome β5 subunit and, to a lesser extent, to the β1 > β2 subunits, with a difference in the β5 subunit dissociation half-life, which is six-fold shorter for IXA than for BTZ [25,29,30,31]. IXA has been approved by the FDA and EMA for the treatment of MM; the drug proved clinically active in relapsed MM [26,32,33,34,35,36]. Moreover, IXA showed in vivo activity in solid tumors and hematological malignancies, including relapsed refractory adult AML [11,37,38]. Whether IXA also holds potential for treatment of pediatric leukemia is relatively unexplored [6]. To gain insight into the mechanisms of action of IXA in leukemia cells, the current study was designed. We employed BTZ-sensitive and -resistant ALL and AML in vitro cell line models and primary ALL and AML clinical specimens to compare the mechanism of action and anti-leukemic activity of IXA vs. BTZ.

## 2. Materials and Methods

### 2.1. Drugs and Materials

IXA and BTZ were kindly provided by Millennium Pharmaceuticals Inc. (Cambridge, MA, USA), a wholly owned subsidiary of Takeda Pharmaceutical Company Limited. Dexamethasone (DEX) was purchased from Sigma-Aldrich (Saint Louis, MO, USA) and cytarabine (Ara-C) from Hospira Benelux BVBA (Brussels, Belgium). Proteasome subunit-specific fluorogenic substrates of β5, β5i, and β1i, AC-WLA-AMC, Ac-ANW-AMC, and Ac-PAL-AMC, respectively, were kindly provided by Millennium Pharmaceuticals Inc. (Cambridge, MA, USA), and the β1 substrate, Z-LLE-AMC, was purchased from BostonBiochem (Cambridge, MA, USA).

### 2.2. Cell Culture

Parental (wild-type (WT)) human myeloid leukemia THP-1 cells, human T-cell lymphatic leukemia CCRF-CEM cells, and human multiple myeloma RPMI-8226 cells (ATCC, Manassas, VA, USA) were cultured in RPMI-1640 medium containing 2 mM L-glutamine (Invitrogen/Gibco, Carlsbad, CA, USA) supplemented with 10% fetal calf serum (FCS; Greiner Bio-one, Alphen a/d Rijn, The Netherlands) and 100 µg/mL penicillin/streptomycin (Invitrogen, Carlsbad, CA, USA) at 37 °C in a 5% CO_2_ humidified atmosphere. Cell cultures were refreshed twice weekly and seeded at a density of 3 × 10^5^ cells/mL. BTZ-resistant cell lines with a low level of resistance (i.e., at 7 nM BTZ) and a high level of resistance (i.e., at 100 or 200 nM BTZ): CEM/BTZ7, CEM/BTZ200, THP-1/BTZ-7, THP-1/BTZ200, 8226/BTZ7, and 8226/BTZ100, were developed and cultured as described previously [39,40].

### 2.3. Patient Samples

Fifteen ALL and nine AML cryopreserved pediatric leukemia samples were included from our pediatric biobank. Disease characteristics of the patient samples are provided in Appendix A. After thawing the vials, viable cells were suspended in RPMI-1640 medium supplemented with 20% FCS and antibiotics and kept overnight in the 37 °C incubator prior to start of the MTT assay.

### 2.4. Proteasome (Subunit) Activity Assay

For measurement of the proteasome activities of β5, β5i, β1, and β1i, subunit-specific fluorogenic substrates were used, with methodologies described in detail before by Niewerth et al. [41]. In short, protein extracts of leukemic cell lines and primary leukemic cells were diluted to 200 µg/mL and plated in individual wells of a 96-well black, opaque, non-binding plate (Greiner Bio-one, Germany) (50 µL). Protein extracts were incubated with a concentration range of IXA or BTZ (in a volume of 20 µL, diluted in 20 mM HEPES, 0.5 mM EDTA, pH 7.4), after which the reactions were initiated by addition of 130 µL 154 µM peptide-AMC substrate (final concentration: 100 µM) at 37 °C. Formation of the fluorescent reaction product was monitored every 5 min at 365 nm on a validated Glomax plate reader (Promega, WI, USA) over a period of two hours. Calculations were based on the linear slopes of the reaction product formation of samples without drugs vs. with drugs. The signal obtained with completely inhibited subunits by excess exposure to 10 µM BTZ served as a background control.

### 2.5. Cell Growth Inhibition Assay

The 4-day 3-[4,5-dimethylthiazol-2-yl]-2,5-diphenyl tetrazolium bromide (MTT) cytotoxicity assay (in triplicate) was used to determine in vitro drug sensitivity, essentially as previously described in detail [40,42]. Prior to this assay, BTZ-sensitive and BTZ-resistant cells were cultured in drug-free medium for at least 1 day. The IC_50_ was defined as the drug concentration that inhibited 50% of the cell growth compared to the untreated control cells (IC_50_). For the drug combination assays, the combination concentrations were based on single drug concentrations, approximating the IC_50_ concentrations of both drugs. Drugs were then combined in the same ratio with higher and lower concentrations of the IC_50_(IXA) + IC_50_(DEX) or IC_50_(IXA) + IC_50_(Ara-C). The combined drugs were diluted in a 12-step dilution scheme and dispensed (in triplicate) into the wells, before adding the cell suspension. Hence, cells were exposed to the drugs simultaneously. The drug combination ranges used are provided in Appendix A. The combination index (CI) was calculated with the use of Calcusyn version 1.1.1 (Biosoft, Cambridge, UK). Calcusyn calculates the CI from the effect of both single drugs with the combined drugs to determine if the effect of both combined drugs is synergistic (CI < 0.8), additive (CI 0.8–1.2), or antagonistic (CI > 1.2) [43,44]. Mean CI was calculated from CIs within a fraction affected (FA) range between 0.5 and 0.9, as recommended by Bijnsdorp et al. [44].

To determine ex vivo drug sensitivity in primary leukemia cells, ALL (2 × 10^6^ cells/mL) or AML (1 × 10^6^ cells/mL) cells were incubated with different concentrations of IXA and BTZ in 100 µL cell culture medium in a 96-well round-bottom plate. Cryopreserved primary cells were carefully thawed and counted for viability by trypan blue staining before adding to the plate (only cell samples with >70% viability were included). After 96 h, before adding MTT, one control well per primary sample was used to examine the quality of the samples with trypan blue count. Patient samples were included when the optical density (OD) values from the MTT-assay were above 0.220. For samples with OD values between 0.120 and 0.220, the following additional criteria were applied: OD values of blank control wells must be at least lower than half of the OD values of the control wells and a regular dose–response curve was obtained. Samples with OD values below 0.120 were excluded. Since these primary patient samples do not proliferate and spontaneous cell death can occur, rather than IC_50_ values, LC_50_ values are presented, referring to the lethal concentrations that result in 50% more cell death compared to untreated control cells [42].

### 2.6. Apoptosis and Cell Cycle Assays (Flow Cytometry)

Cells were seeded at 5 × 10^5^/mL in a 6-well plate and exposed to IXA concentrations (range: 5–10,000 nM) for 24 h. IXA-induced apoptosis was determined by measuring 7-amino-actinomycin D (7-AAD) (BD Via-Probe™, BD Bioscience, San Jose, CA, USA) and Annexin V-FTIC staining (Apoptest™–FITC A700, VPS Diagnostics, Hoeven, The Netherlands) according to manufacturer’s instructions. Cells were first incubated for 15 min with 7-AAD and then washed with phosphate buffered saline (PBS) containing 0.1% human serum albumin before Annexin-V staining. Cells were measured using a Fortessa flow cytometer (BD Bioscience) and analyzed with FACSDiva software (BD Bioscience).

Cells seeded for apoptosis assay were also used for cell cycle analysis. Cells were washed with PBS, fixed with 70% ice-cold ethanol, and placed at 4 °C overnight. After washing the cells twice with PBS, cells were incubated with RNAse (100 µg/mL) for 30 min at 37 °C. Then, propidium iodide was added at a concentration of 50 µg/mL, followed by analysis (25,000 events each condition) with a Fortessa flow cytometer.

### 2.7. Statistics

For the proteasome activity assay, slopes were calculated using a linear regression analysis. Paired t-testing was used to determine the statistical significance of the differences between IXA and BTZ. Statistical significance between AML and ALL was determined using the Mann–Whitney U test. Statistical significance was achieved when *p* < 0.05. All statistical analyses were performed using GraphPad Prism Version 8.2 software (La Jolla, CA, USA).

## 3. Results

### 3.1. Mechanism of Action of IXA and BTZ: Subunit Inhibition Profile in (BTZ-Resistant) Leukemia Cell Lines

The inhibition profile by IXA vs. BTZ of two constitutive proteasome (cP) subunits β5 and β1 and their immunoproteasome (iP) counterparts β5i and β1i was examined in CEM (ALL) and THP-1 (AML) WT cells and their BTZ7 (low level of BTZ resistance) and BTZ200 (high level of BTZ resistance) sublines (Figure 1). For WT cells, inhibition of β5 and β5i was slightly more potent by BTZ compared to IXA. However, β1 inhibition, and, in particular, β1i inhibition, was more potent by IXA than BTZ—e.g., in CEM/WT, at 5 nM IXA, β1i inhibition was 90% vs. 63% inhibition by 5 nM BTZ (*p* = 0.01). For β5i, β1, and β1i, these inhibition profiles were comparable in BTZ7 and BTZ200 cells, but for β5, increasingly higher concentrations of both BTZ and IXA were required for (50%) inhibition. This is explained by the upregulation of β5 subunit expression in BTZ-resistant cells [40], which is also reflected by increased absolute β5 activity (increased slopes) in BTZ-resistant vs. WT cells (Appendix A).

### 3.2. Sensitivity of BTZ-Sensitive and BTZ-Resistant ALL, AML, and MM Cell Lines to IXA

The in vitro anti-leukemic activity of IXA was determined in CEM and THP-1 cells and, for comparison, 8226 (MM) cells and their BTZ-resistant sublines with a low level of BTZ resistance (/BTZ7) and a high level of BTZ resistance (/BTZ200 and /BTZ100, respectively). The IXA dose–response curves for 4-day cell growth inhibition assays are depicted in Figure 2 and IC_50_ values are presented in Table 1. For all three WT cell lines, it is noteworthy that, concentration-wise, BTZ is ~10-fold more potent than IXA (Table 1), although IC_50_ values for IXA were still in the low nanomolar range. All six BTZ-resistant cell lines displayed cross-resistance to IXA with a factor of 5–11 in the /BTZ7 cells (being 5–27-fold resistant to BTZ) and a factor of 103–111 in CEM/BTZ200 and THP-1/BTZ200 cells and 8226/BTZ100 cells (being 40–170-fold resistant to BTZ).

### 3.3. IXA-Induced Apoptosis and Effects on Cell Cycle in Leukemia Cells

Exposure of CEM/WT cells to 20 nM IXA for 24 h resulted in considerable apoptosis induction (Figure 3A). Quantification of apoptosis induction for CEM/WT and THP-1/WT over a range of 20–100 nM IXA exposure for 24 h showed percentages of apoptotic cells of 43–94% and 39–83%, respectively (Figure 3B). In addition, 24 h IXA exposure also had an impact on the cell cycle of CEM and THP-1 cells and their low/high BTZ-resistant sublines with an increase in G2/M phase, which is most pronounced in CEM/WT (19%) and THP-1/BTZ7 cells (35%) (Figure 3C).

### 3.4. Combination Effects of IXA with DEX and Ara-C

Since (pediatric) acute leukemia patients will not be treated with IXA as monotherapy, IXA was also tested in combination with chemotherapeutics commonly used for the treatment of ALL and AML (Table 2). For ALL, IXA was tested in combination with DEX in the CEM cell line and its BTZ-resistant sublines. The combination of IXA and DEX showed a moderate synergistic effect in the low BTZ-resistant CEM cell line (mean CI: 0.77 ± 0.3). In contrast, in the CEM/WT and high BTZ-resistant cell lines, a moderate additive-to-antagonistic effect was found with a CI of 1.22 ± 0.1 and 1.31 ± 0.2, respectively. For the treatment of AML, Ara-C is commonly used and was, therefore, tested in combination with IXA. Notably, this combination only showed antagonistic effects in the THP-1 cell line, with CI between 1.62 and 2.39 (Table 2). The IC_50_ values of the single drugs are depicted in Appendix A.

### 3.5. Sensitivity of Primary ALL and AML Samples to IXA vs. BTZ

Beyond analysis of IXA sensitivity in leukemia cell lines (Figure 2), we also assessed the sensitivity in primary ALL and AML patient samples. MTT assays were performed on 15 pediatric ALL and 9 AML leukemia samples (including two relapsed ALL patients) to investigate the ex vivo anti-leukemic effects of IXA vs. BTZ (Figure 4). LC_50_ values for the individual samples are presented in Figure 4A. ALL and AML cells were significantly more sensitive to BTZ than IXA, with ALL mean LC_50_ values of 4.5 ± 1 nM for BTZ and 24 ± 11 nM for IXA (*p* = <0.0001), and for AML, mean LC_50_ values of 11 ± 4 nM for BTZ and 30 ± 8 nM for IXA (*p* = 0.0001). In line with our previous data [18], ALL cells were significantly more sensitive than AML cells to BTZ (*p* = <0.0001). However, no significant difference was found between ALL and AML cells for IXA (*p* = 0.0813). The combined dose–response curves for IXA vs. BTZ sensitivity in ALL and AML samples are shown in Figure 4B. Next, we examined whether the LC_50_ values for IXA and BTZ were correlated (Figure 4C). In ALL samples, increasing LC_50_ values for BTZ were correlated (*p* = 0.027, r = 0.57). Interestingly, in AML samples, LC_50_ values for BTZ and IXA were not correlated (*p* = 0.74, r = 0.13) and cells with lower sensitivities for BTZ remained sensitive to IXA.

## 4. Discussion

The pivotal role of the ubiquitin–proteasome system for protein degradation and processing in immune cells has been successfully exploited for the treatment of hematological malignancies such as MM by a PI, initially BTZ [12,46,47,48,49]. The present study was undertaken as a first step to pre-clinically evaluate whether, beyond MM treatment, IXA could hold potential anti-leukemic effectivity. To this end, proteasome inhibition profiling and assessment of anti-leukemic activity of IXA were performed and compared with BTZ for leukemia cell lines, BTZ-resistant sublines, and primary samples of acute leukemia patients.

IXA showed overlapping activities with BTZ, although equipotency in T-ALL and AML cell lines was reached at approximately 10-fold higher concentrations than BTZ in the BTZ-sensitive and -resistant cell lines. Moreover, IXA was a more potent inhibitor of β1i immunoproteasome subunits than BTZ. However, this property did not overcome cross-resistance to IXA in BTZ-resistant cell lines.

To anticipate the potential efficacy of PIs in leukemia cells, understanding of the proteasome subunit composition is of relevance, as unlike solid tumors, the dominant fraction (>70%) is composed of iP subunits rather than cP subunits [6,11,41,50]. In fact, a higher ratio of iP/cP has been proposed as an indicator for ex vivo BTZ sensitivity in primary leukemia cells [18,51]. The IXA cP inhibition profile in leukemia cells confirms earlier studies’ results in MM cell lines that IXA is a potent inhibitor of β5 activity and, to a lesser extent, β1 activity [25,29]. The current study shows that IXA is an even more potent inhibitor of iP subunits, β1i activity, as well as β5i activity compared to cP subunits. It is not clear whether inhibition of these iPs contributes to the growth inhibitory effects of IXA, as retention/dissociation time on the β1i subunit is unknown. Earlier studies with selective iP inhibitors indicated that, primarily, inhibition of the β5 subunit confers anti-leukemic effects [6,50,52]. This notion is consistent with upregulated β5 expression in BTZ-resistant cells, which explains the higher concentrations of IXA and BTZ required to inhibit β5 in /BTZ7 and, to a greater extent, /BTZ200 cells [40,41,52]. In this regard, it should be taken into account that the leukemia cell lines, CEM and THP-1, used in this study may not fully represent the cP vs. iP distribution in primary cells, as in these in vitro models, cP expression is two-fold higher than iP expression [41]. Furthermore, although we used well-characterized BTZ-resistant sublines of cell lines representative for ALL (CEM) and AML (THP-1), this also sets a limitation to this study, and confirmatory studies for a broader panel of (BTZ-resistant) ALL and AML cells are recommended, particularly for a heterogenous malignancy such as AML. In this respect, recent studies revealed that nucleophosmin *NPM1* mutation status in AML cells was associated with IXA sensitivity [53,54], indicating that the response to IXA may be genetic context-dependent.

Direct comparison of the growth inhibitory effectiveness of IXA vs. BTZ in leukemic cells lines showed that BTZ was ~10-fold more potent than IXA, being consistent with data for MM cell line studies [29]. Conceivably, this is explained by the much shorter dissociation half-life (18 vs. 110 min) of IXA at the β5 subunit than BTZ [25]. The involvement of β5 subunit inhibition for IXA effectiveness is further corroborated by IXA cross-resistance in BTZ-resistant leukemia cells. These cell lines acquired BTZ resistance by point mutations in exon 2 of the *PSMB5* gene, encoding the β5 subunit, introducing amino acid substitutions (Thr21Ala, Met45Ile, Ala49Thr, and Cys52Phe) in the conserved BTZ binding pocket of the β5 subunit protein [39,40]. Exon 2 *PSMB5* mutations have been commonly found in hematological and solid tumor cell line models [55,56,57,58,59,60] and, most recently, also in subclones of malignant cells of MM patients receiving BTZ therapy [22]. Another recent study by Brunnert et al. [61] also identified *PSMB5* mutation in MM cell line models with acquired resistance to IXA, which were cross-resistant to BTZ. Interestingly, these *PSMB5* mutations in IXA-resistant MM cells introduced an additional amino acid substitution (Ala50Val), indicating another critical amino acid in the β5 binding pocket for IXA. This may not be surprising given the two bulky chlorine moieties in the chemical structure of IXA [28]. Although IXA and BTZ may share *PSMB5* mutation as a common molecular mechanism of (cross-)resistance, it is important to acknowledge that several other mechanisms of resistance have been reported [21,48,62] which need further investigation for their relevance for IXA contributing to overcome BTZ resistance and for onset of resistance to IXA itself. These include inherent factors impacting redox homeostasis, glycolytic rates, pro-survival signaling pathway activities (MAPK, PI3K/AKT, NFκB), and acquired factors, e.g., increased autophagy, impaired apoptosis, reduced XBP1 expression, and upregulation of MARCKS [63,64,65,66,67]. Upregulation of ABC drug efflux transporters is not implicated in acquired resistance for IXA, nor BTZ [61,68].

Even though primary ALL and AML cells differ from leukemia cell lines in the constitutive and immunoproteasome composition profile [18,41,51], the effect of IXA on cell viability in primary ALL and AML leukemia samples was in the low nanomolar range. BTZ was significantly more potent than IXA in ALL cells, while this difference was smaller (not significant) for AML cells (Figure 4). Extended research, including pre-clinical animal models and larger sets of leukemia patient samples, is warranted to fully define the clinical potential of IXA for the treatment of pediatric acute leukemia. These further studies should also be designed to correlate IXA sensitivity to the molecular genomic profile of ALL and AML samples; these analyses were not feasible in the present study due to the low sample size.

The lower (in vitro) potency of IXA compared to BTZ may, in vivo, be compensated by a better pharmacokinetic profile with higher plasma levels and longer plasma half-life of IXA [25,26,69]. Orally administered IXA is rapidly absorbed and shows systemic exposure similar to intravenous administration [35,70]. Moreover, the distribution of IXA from blood into tissues in pre-clinical experiments is five times higher compared to BTZ [34]. Importantly, IXA has less of an effect on the dorsal root ganglion of the spinal cord, which is responsible for PN during BTZ therapies [71,72,73].

Considering that IXA will not be used as a single agent in leukemia treatment, it is important to identify potential candidates among commonly used anti-leukemic drugs for combination therapy regimens in pediatric acute leukemia treatment. In this respect, a study in relapsed/refractory AML revealed that IXA did not appear to increase the toxicity of the standard MEC regimen with mitoxantrone, etoposide, and Ara-C, although effectivity was limited [38]. In MM, IXA showed limited additional toxicity and good tolerability in combination regimens, which include cyclophosphamide, dexamethasone, thalidomide, melphalan, and prednisone [32]. Given their non-overlapping mechanisms of action, combinations of IXA and glucocorticoids (e.g., dexamethasone) deserve further exploration in acute leukemia. Pre-clinical combination studies in pediatric ALL samples and in vitro studies with BTZ and other anti-leukemic drugs showed cell line-dependent variability in levels of synergy, additivity, and antagonism [18,74]. Moreover, clinical studies of BTZ in children with relapsed and refractory ALL showed promising results with combination regimens including mitoxantrone, vincristine, asparaginase, doxorubicin, and dexamethasone [14,15,16,17]. In vitro studies in AML cells with IXA in combination with Ara-C showed increased chemosensitivity [54]. Moreover, synergistic effects were observed for IXA in combination with the HDAC inhibitor SAHA in NPM_C_^+^ AML cells [53]. More recently, one clinical trial with IXA in combination with mitoxantrone, etoposide, and Ara-C in adult AML showed moderately increased chemosensitivity [38]. Our observation of IXA sensitivity in primary AML cells (Figure 4) underscores these observations. However, our initial combination experiments of leukemia cells with IXA and DEX or Ara-C in the present study did not show consistent additive/synergistic effects and even antagonistic effects. These antagonistic effects may be explained by the fact that, when added simultaneously, IXA causes cell cycle arrest whereas Ara-C is most effective in cycling cells. Additional studies with different time spans and several sequences of drug administration are advocated to more thoroughly examine the combined effect of IXA and other anti-leukemic drugs to position the most effective drug combination schedules.

In conclusion, IXA has overlapping mechanisms of action with BTZ and showed anti-leukemic activity (in the low nanomolar range) in (CEM and THP-1) leukemia cell lines in vitro and in primary leukemic cells, including AML and relapsed ALL patient samples. However, for equi-effective activity, higher concentrations of IXA were needed than for BTZ in ALL. This apparent shortcoming may be compensated by IXA’s more favorable pharmacokinetic and safety profile over BTZ, as demonstrated in clinical studies in MM and adult acute leukemia. Therefore, optimal IXA dose scheduling deserves further exploration in future pre-clinical leukemia models.

## Figures and Tables

**Figure 1 cells-10-00665-f001:**
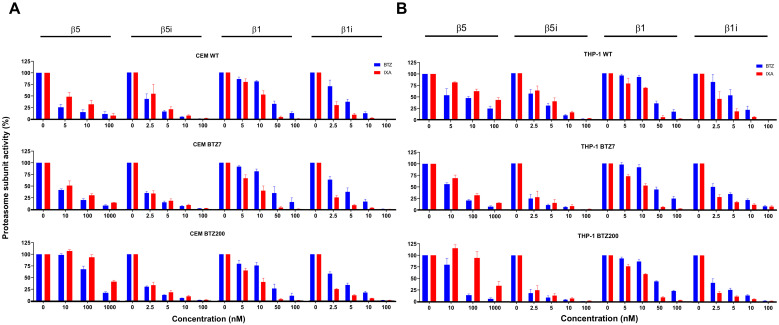
Ixazomib (IXA) vs. Bortezomib (BTZ) inhibition profile of proteasome subunit activity in CEM (acute lymphoblastic leukemia, ALL) and THP-1 (acute myeloid leukemia, AML) cells and their low/high BTZ-resistant sublines. β5-, β5i-, β1-, and β1i-associated catalytic activity in CEM (**A**) and THP-1 (**B**) cell extracts of wild-type (WT), BTZ7 (low level BTZ resistance), and BTZ200 (high level BTZ resistance) was assessed in the absence (control) or presence of increasing concentrations of IXA or BTZ. Results depicted represent the mean ± SEM of three separate experiments.

**Figure 2 cells-10-00665-f002:**
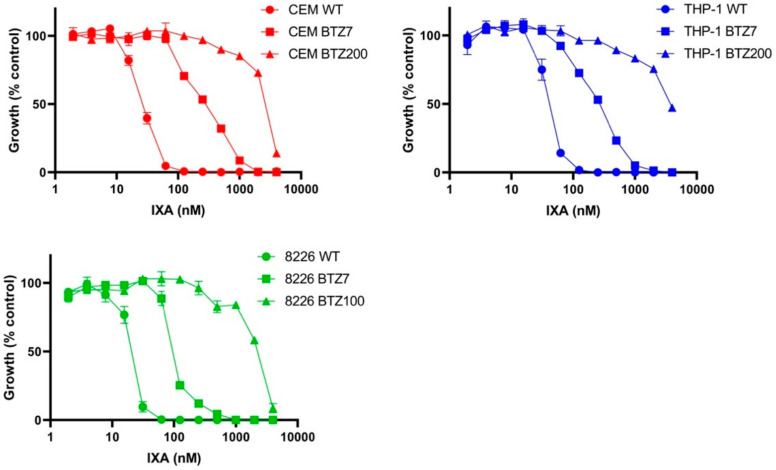
IXA sensitivity of BTZ-sensitive and BTZ-resistant cells. Sensitivity for IXA was assessed after 4-day drug exposure by the MTT cytotoxicity assay in CEM and THP-1 (8226 for comparison^42^) cells and their low (/BTZ7) and high (/BTZ200 and /BTZ100) BTZ-resistant sublines. Results are expressed as cell growth relative to control cells incubated without drug, set at 100%. Dose–response curves depict the mean ± SEM of three separate experiments.

**Figure 3 cells-10-00665-f003:**
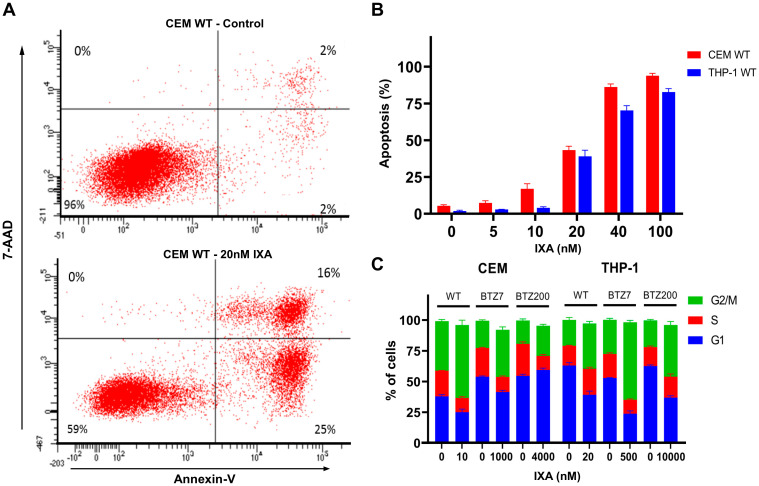
IXA-induced apoptosis and effects on cell cycle in (BTZ-resistant) leukemia cells. (**A**) Representative flow cytometry image of apoptosis induction (Annexin-V and 7-amino-actinomycin D (7-AAD) staining) for control CEM/WT cells (upper left) and CEM/WT cells exposed for 24 h to 20 nM IXA (lower left). (**B**) Quantification of apoptosis induction in CEM/WT and THP-1 cells exposed for 24 h to 0–100 nM IXA. Results show percentages of apoptotic cells ± SEM of three separate experiments. (**C**) Cell cycle analysis profile (propidium iodide staining) of CEM/WT and THP-1/WT cells and their low/high BTZ-resistant sublines. Cells were exposed for 24 h to the indicated concentrations of IXA. Results: Distribution of cells over G0, S, and G2/M phases is depicted as the mean ± SEM of three separate experiments.

**Figure 4 cells-10-00665-f004:**
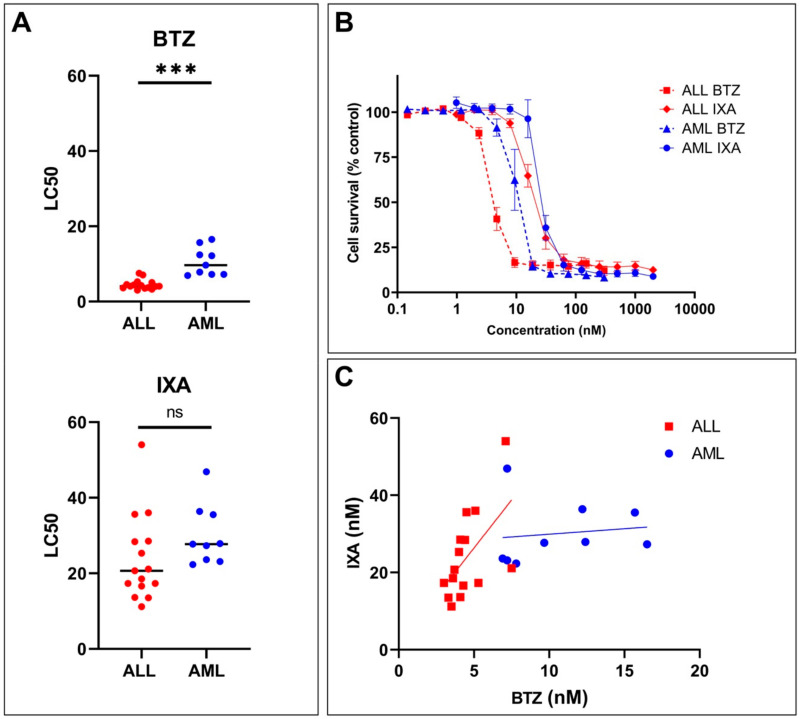
Sensitivity of primary pediatric (relapsed) ALL and AML patient samples to IXA and BTZ. (**A**) Individual LC_50_ values (nM) for IXA and BTZ as obtained after 4-day drug exposure and MTT cytotoxicity assay analysis in ALL and AML patient samples. The line indicates the mean LC_50_ value (nM), *** indicates *p* < 0.0001. (**B**) Combined dose–response curves of cell survival (relative to control set at 100%) of ALL and AML cells exposed for 4 days to IXA and BTZ. Results present the mean cell survival at each drug concentration ± SEM. (**C**) Correlation of the LC_50_ values (nM), from Figure 4A, for IXA and BTZ in ALL and AML patient samples.

**Table 1 cells-10-00665-t001:** Growth inhibitory effects of IXA compared to BTZ in hematological cell lines and their BTZ-resistant sublines.

	IC_50_ IXA ± SD, nM (RF)	IC_50_ BTZ ± SD, nM (RF) *
**Wild-type cell lines**		
CEM	27 ± 2	1.5 ± 0.4
THP-1	38 ± 9	2.6 ± 0.6
8226 ^#^	22 ± 2	2.6 ± 0.3
**BTZ low resistance sublines**		
CEM/BTZ7	291 ± 29 (11)	12.4 ± 5.8 (10)
THP-1/BTZ7	277 ± 26 (7)	70 ± 10 (27)
8226/BTZ7 ^#^	100 ± 1 (5)	12.1 ± 0.7 (5)
**BTZ high resistance sublines**		
CEM/BTZ200	2784 ± 31 (103)	189 ± 44 (170)
THP-1/BTZ200	3817 ± 31 (103)	390 ± 68 (153)
8226/BTZ100 ^#^	2332 ± 105 (106)	106 ± 15 (40)

Results are expressed as cell growth relative to control cells incubated without drug, set at 100%. Mean ± SD of three separate experiments performed in triplicate. Abbreviations: SD, standard deviation; RF, resistance factor; * adapted from Franke et al. and Oerlemans et al. [39,40]; ^#^ adapted from Zweegman et al. [45].

**Table 2 cells-10-00665-t002:** Mean combination index (CI) (± SD) of IXA combined with dexamethasone (DEX) and cytarabine (Ara-C) in BTZ-sensitive and BTZ-resistant leukemia cells.

	Combination Index DEX	Combination Index Ara-C
**T-ALL cells**		
CEM/WT	1.22 ± 0.1	-
CEM/BTZ7	0.77 ± 0.3	-
CEM/BTZ200	1.31 ± 0.2	-
**AML cells**		
THP-1/WT	-	2.16 ± 0.3
THP-1/BTZ7	-	2.39 ± 1.8
THP-1/BTZ200	-	1.62 ± 0.3

## Data Availability

Data are available upon request from the corresponding author.

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
