# Peer review of "Pre-Clinical Evaluation of the Proteasome Inhibitor Ixazomib against Bortezomib-Resistant Leukemia Cells and Primary Acute Leukemia Cells"

_cells, 2021, doi:10.3390/cells10030665_

Round 1

Reviewer 1 Report

In this manuscript the authors report in vitro activity of a newer proteasome inhibitor, ixazomib, in models of ALL and AML with resistance to bortezomib. Ixazomib is potentially better than bortezomib as it is orally bioavailable, has lower risk of peripheral neuropathy, and a longer half-life. They studied one AML cell line, THP-1, one T-ALL cell line (CEM) and one MM cell line. Each cell line included WT and bortezomib-resistant sublines. They also studied a decent number of pediatric patient samples, both ALL and AML. They report that ixazomib is more active against the B1 subunit than B5, unlike bortezomib. Ixazomib induces cytotoxicity (as measured by MTT assay) and apoptosis (Annexin V/7-AAD) of these cell lines, and resistance to bortezomib also confers resistance to ixazomib. The paper is well written and the figures are easy to understand. The conclusions that can be drawn are limited due to the fact that only one cell line of each major diagnostic category was studied. Also, there are key details for some of the methods that are missing, making it difficult to evaluate some of the results.  Specific points follow.

  1. A significant limitation of the study is the few number of cell lines studied. Ideally the study would include several ALL and AML cell lines. If not, the authors should avoid overstating conclusions about ALL or AML in general, and they should acknowledge this limitation in the discussion.
  1. Regarding the combination drug testing, please describe in more detail how the experiment was done. Specifically, were the two drugs added together or sequentially? The sequence and timing of combination treatment can profoundly affect how two drugs interact. Since ixazomib causes cell cycle arrest and cytarabine is most effective in cycling cells, it is not at all surprising that they are antagonistic if given together or with ixazomib first.
  1. Please also describe in more detail how the dose response testing was done for the primary samples. Specifically, how was spontaneous cell death, which can be quite high, accounted for? Four days is a long time to culture primary cells without some kind of cytokine support. What were the MTT values of untreated cells after 4 days in culture? What criteria were used to determine if a sample was healthy enough for valid drug response testing?

Reviewer 2 Report

The study presents data on the activity of the proteasome inhibitor ixazomib in comparison to bortezomib against primary ALL or AML cells and against bortezomib resistant cell lines. There are some differences in the inhibitory potential against proteasome subunits activity and growth inhibition, as well as in the lethal concentrations. The authors conclude that these findings warrant further pre-clinical and in vivo investigation of IXA. 

However, research on IXA is ongoing and clinical studies on combination therapies including IXA for the therapy of relapsed refractory acute myeloid leukemia are published (CCR, Advani et al., 2019). This study is (shortly) mentioned in the discussion but should be presented in the introduction. There are also several in vitro studies on IXA activity against leukemia cells, which should be mentioned (e.g. JCI Insight 2018, Kahn et al.; CCR 2016, Garcia et al.).

Author Response

Response to Reviewer 2 Comments

General remarks:

The study presents data on the activity of the proteasome inhibitor ixazomib in comparison to bortezomib against primary ALL or AML cells and against bortezomib resistant cell lines. There are some differences in the inhibitory potential against proteasome subunits activity and growth inhibition, as well as in the lethal concentrations. The authors conclude that these findings warrant further pre-clinical and in vivo investigation of IXA. 

However, research on IXA is ongoing and clinical studies on combination therapies including IXA for the therapy of relapsed refractory acute myeloid leukemia are published (CCR, Advani et al., 2019). This study is (shortly) mentioned in the discussion but should be presented in the introduction. There are also several in vitro studies on IXA activity against leukemia cells, which should be mentioned (e.g. JCI Insight 2018, Kahn et al.; CCR 2016, Garcia et al.).

Reply: We thank the reviewer for acknowledging that this study harbors relevant preclinical information, but invites the inclusion of some additional (pre)clinical studies addressing IXA research in leukemia cells and a clinical study in adult AML. To meet the reviewers point we now cited the clinical study with IXA in relapsed refractory adult AML by Advani et al [ref 38] in the Introduction section (line 73-74).

Next to this, we referred to the IXA in vitro studies by Garcia et al [ref 53] and Kahn et al [ref 54] in the Discussion section (lines 310-312 and 369-371) and indicated the relevance of NPM1 status in AML cells for IXA response.

References
1. Garcia, J.S.; Huang, M.; Medeiros, B.C.; Mitchell, B.S. Selective Toxicity of Investigational
Ixazomib for Human Leukemia Cells Expressing Mutant Cytoplasmic NPM1: Role of Reactive
Oxygen Species. Clin. Cancer Res. 2016, 22, 1978–1988, doi:10.1158/1078-0432.CCR-15-1440.
2. van Meerloo, J.; Kaspers, G.J.; Cloos, J. Cell sensitivity assays: the MTT assay. Methods Mol
Biol 2011, 731, 237–245, doi:10.1007/978-1-61779-080-5_20.

Round 2

Reviewer 1 Report

Thank you for the reply and revisions to the manuscript.

This manuscript is a resubmission of an earlier submission. The following is a list of the peer review reports and author responses from that submission.

Round 1

Reviewer 1 Report

The study compare the in vitro proteasome  inhibition profile of IXA vs BTZ against leukemia cell Lines, BTZ- resistant sublines and primary samples of ALL and AML patients( pediatric).
Some useful  informations  are  given:
- IXA showed overlapping activity with BTZ and resulted to inhibit Beta1i subunit, while BTZ the Beta5 subunit.
- IXA resulted to be  10 fold lesser potent than BTZ and moreover IXA did not show to overcome cross-resistance in BTX resistant cell lines
The Authors speculate that the lower inhibitory effects of IXA against ALL/AML cells could  Be favorably counterbalanced by the better toxicity profile
My be, but the Authors should give more important data
- They should comparatively evaluate the in vitro effects of IXA and BTZ in combination with the drugs commonly used in ALL/AML resistant/relapsed patients, taking into account that BTZ has been already tested in vivo.
If a synergistic and better Or equal effects  of IXA in combinations with cytotoxic drugs vs BTZ combinations would result, the potential in vivo use of IXA could Be considered
- The inhibitory effects of IXA and BTZ appear to be lesser intensive in AML than in ALL . A correlation between the IXA and BTZ activities and the molecular/genomic profile of AML/ALL cell samples would be warranted.

Author Response

Response to Reviewer 1 Comments

As a general reply to the reviewers we would like to emphasize that this manuscript was written in the context of a special issue of Cells with the topic of studying drug resistance using cancer cell lines. We have employed ALL and AML cell lines with and without acquired resistance to BTZ as reference to examine the activity of the new proteasome inhibitor IXA, which is clinically evaluated in multiple myeloma, but not yet in a (pediatric) acute leukemia setting. The results of this study show that BTZ and IXA share both overlapping and different mechanisms of action in (BTZ-resistant) acute leukemia cells and display ex vivo activity in clinical specimen of ALL and AML. These observations should be helpful in the accurate design of follow-up preclinical (animal) studies and potential future clinical testing of IXA in acute leukemia, studies being beyond the scope of the present manuscript.      

General remarks:

The study compare the in vitro proteasome inhibition profile of IXA vs BTZ against leukemia cell lines. BTZ resistant sublines and primary samples of ALL and AML patient (pediatric). Some useful informations are given:

- IXA showed overlapping activity with BTZ and resulted to inhibit Beta1i subunit, while BTZ the Beta5 subunit:

- IXA resulted to be 10 fold lesser potent than BTZ and moreover IXA did not show to overcome cross-resistance in BTZ resistant cell lines. The authors speculate that the lower inhibitory effect of IXA against ALL/AML cells could be favorably counterbalanced by the better toxicity profile. May be, but the authors should give more data

Reply: We thank the reviewer for indicating that this study present useful information, but invites for presenting more data. In the revised manuscript we added a Table presenting the results of IXA-drug combination testing with other leukemia drugs for (BTZ-resistant) ALL and AML cells lines (see also reply to specific remark 1 below). 

Specific remarks:

  1. They should comparatively evaluate the in vitro effects of IXA and BTZ in combination with the drugs commonly used in ALL/AML resistant/relapsed patients, taking into account that BTZ has been already tested in vivo. If a synergistic and better Or equal effects of IXA in combinations with cytotoxic drugs vs BTZ combinations would result, the potential in vivo use of IXA could be considered.

Reply 1: The reviewer raises a valid point. In the revised manuscript (lines 222-234) we added an additional paragraph and Table (Table 2) presenting the combinations indexes (CI) for IXA/BTZ combined with either dexamethasone and cytarabine for (BTZ-resistant) ALL and AML cell lines in vitro. No clear synergistic effects were observed for IXA/BTZ combined with dexamethasone and cytarabine.

  1. The inhibitory effects of IXA and BTZ appear to be lesser intensive in AML than ALL. A correlation between the IXA and BTZ activities and the molecular genomic profile of AML/ALL cell samples would be warranted.

Reply 2: We had previously shown (Niewerth et al, Haematologica 2013;98:1896-1904) that ex vivo sensitivity of primary ALL cells for BTZ is greater than AML cells. Such a profile is also commonly observed for other anti-leukemic drugs in ALL vs AML cells (Kaspers et al, Leukemia 1994; 8:1224-1229). The results for IXA are consistent with these notions. We agree with the reviewer that assessments/correlations of IXA vs BTZ activity warrant inclusion of molecular genomic profiling. Unfortunately, for the limited number of clinical samples used in the ex vivo testing, cryopreserved samples were used which had not been subjected to extensive molecular profiling as currently done, but rather solely on the basis of immunophenotyping/FAB classification (for AML). We indeed would advocate that in future clinical studies with larger numbers of leukemia patients, molecular profiling is included for BTZ/IXA activity correlation analyses. We added a statement on this (lines 312-314) to acknowledge the reviewer’s remark on this point.    

Reviewer 2 Report

Authors show a new proteasome inhibitor that is not as potent as the currently used PI using In vitro assay on the leukemia cells. Minimal original data and findings presented here do not substantiate presentation in the form of a research article. The results show that the new PI inhibitor (IXA) is far less potent compared to BTZ. Further testing in preclinical leukemia models to characterize  PD and efficacy profile is essential to support the author's conclusions. In vitro cytotoxicity results alone is not sufficient. In-depth evaluation of the novel compound in at least cell line derived in vivo models would increase the significance of content and interest readers.

Author Response

As a general reply to the reviewers we would like to emphasize that this manuscript was written in the context of a special issue of Cells with the topic of studying drug resistance using cancer cell lines. We have employed ALL and AML cell lines with and without acquired resistance to BTZ as reference to examine the activity of the new proteasome inhibitor IXA, which is clinically evaluated in multiple myeloma, but not yet in a (pediatric) acute leukemia setting. The results of this study show that BTZ and IXA share both overlapping and different mechanisms of action in (BTZ-resistant) acute leukemia cells and display ex vivo activity in clinical specimen of ALL and AML. These observations should be helpful in the accurate design of follow-up preclinical (animal) studies and potential future clinical testing of IXA in acute leukemia, studies being beyond the scope of the present manuscript.      

General remarks:

Authors show a new proteasome inhibitor that is not as potent as the currently used PI using in vitro assay in the leukemia cells. Minimal original data and findings presented here do not substantiate presentation in the form of a research article. The results show that the new PI inhibitor (IXA) is far less potent compared to BTZ.

Reply: We respectfully would like to refer to the remark made above regarding the context of this research article for this special issue of Cells. Moreover, we would like to indicate that preclinical research on IXA has been mostly focused on multiple myeloma and only to a limited extent on acute leukemia and BTZ-resistant sublines thereof. As such, the present study does convey original data for IXA in leukemia, regardless of overlapping mechanisms of action with multiple myeloma. Indeed, concentration-wise, IXA showed a lower in vitro and ex vivo potency than BTZ (still in the lower nM range) in acute leukemia cells. However, this does not necessarily imply that IXA would be inferior to BTZ in vivo. Given its better safety profile, IXA may be eligible for higher dosing and alternative schedules than BTZ. These warrants further examination in additional (pre)clinical studies which go beyond the scope of the present study.    

Specific remarks:

Further testing in preclinical leukemia models to characterize PD and efficacy profile is essential to support the author’s conclusions. In vitro cytotoxicity results alone is not sufficient. In-depth evaluation of the novel compound in at least cell line derived in vivo models would increase the significance of content and interest readers.

Reply: As indicated above, we acknowledge that additional preclinical (animal leukemia models) studies and clinical studies will be required to fully define that potential of IXA as anti-leukemic drug. We recognize that we may have over-stated the potential of IXA in the Discussion section of the manuscript. Therefore, to meet the reviewer’s remark, we moderated these statements by referring that preclinical leukemia models, PD and (in vivo) efficacy assessments are warranted to reveal the anti-leukemic potential of IXA (Discussion section: lines 310-313).

Round 2

Reviewer 1 Report

The new experiments suggested give further information on the in vitro activity of IXA against BTZ resistant cell lines and ALL/AML leukemic cells either alone and in combination with conventional drugs DX or AraC

The results showed neither synerginestic nor additive effects of IXA in combination with DX/AraC, rather than antagonistic effects.

These data suggest that IXA or BTZ, being proteasome inhibitors, should be preclinically tested with other biological compounds addressing molecular targerts of pathways involved in the biology of ALL/AML 

Reviewer 2 Report

Authors suggest that IXA has better tolerability in early human clinical trials which makes less potent cytotoxic effect even less relevant. BTZ has not shown promising results in clinical setting in pediatric leukemia. Again there is minimal original data to merit publication as  Resistance mechanism driving PIs resistance is not well discussed. Further evaluation of mechanism of resistance and how IXA may help overcome resistance to BTX would strengthen the paper. Proteasome subunit assay of IXA treated primary cells is relevant . Authors may need to pursue more in depth investigation to strengthen the manuscript for Cells.

Minor: The molecular genetic characteristics of primary leukemia cells are not mentioned.